# Single-particle cryo-EM structure of a voltage-activated potassium channel in lipid nanodiscs

Doreen Matthies[1†], Chanhyung Bae[2†], Gilman ES Toombes[2], Tara Fox[3,4], Alberto Bartesaghi[1], Sriram Subramaniam[1]*, Kenton Jon Swartz[2]*

[1]Laboratory of Cell Biology, Center for Cancer Research, National Cancer Institute, National Institutes of Health, Bethesda, United States; [2]Molecular Physiology and Biophysics Section, Porter Neuroscience Research Center, National Institute of Neurological Disorders and Stroke, National Institutes of Health, Bethesda, United States; [3]Center for Molecular Microscopy, Center for Cancer Research, National Cancer Institute, National Institutes of Health, Bethesda, United States; [4]Cancer Research Technology Program, Frederick National Laboratory for Cancer Research, Frederick, United States

*For correspondence:
subramas@mail.nih.gov (SS);
swartzk@ninds.nih.gov (KJS)

[†]These authors contributed equally to this work

**Abstract** Voltage-activated potassium (Kv) channels open to conduct $K^+$ ions in response to membrane depolarization, and subsequently enter non-conducting states through distinct mechanisms of inactivation. X-ray structures of detergent-solubilized Kv channels appear to have captured an open state even though a non-conducting C-type inactivated state would predominate in membranes in the absence of a transmembrane voltage. However, structures for a voltage-activated ion channel in a lipid bilayer environment have not yet been reported. Here we report the structure of the Kv1.2–2.1 paddle chimera channel reconstituted into lipid nanodiscs using single-particle cryo-electron microscopy. At a resolution of ~3 Å for the cytosolic domain and ~4 Å for the transmembrane domain, the structure determined in nanodiscs is similar to the previously determined X-ray structure. Our findings show that large differences in structure between detergent and lipid bilayer environments are unlikely, and enable us to propose possible structural mechanisms for C-type inactivation.
DOI: https://doi.org/10.7554/eLife.37558.001

## Introduction

Voltage-activated potassium (Kv) channels open in response to membrane depolarization to conduct $K^+$ ions, a process that is critical for electrical signaling in neurons and endocrine cells, as well as muscle contraction (*Bocksteins, 2016*; *Jan and Jan, 2012*). Kv channels contain voltage-sensing domains that activate with membrane depolarization, and a central pore domain that opens to conduct $K^+$ ions with high selectivity (*Bezanilla, 2008*; *Swartz, 2008*). In the continual presence of membrane depolarization, most Kv channels subsequently enter non-conducting states through distinct mechanisms of inactivation (*Hoshi et al., 1991*; *Yellen, 2002*). In the Shaker Kv channel, for example, opening of the channel in response to membrane depolarization is only transient because the channel rapidly (msec) enters the N-type inactivated state (*Timpe et al., 1988*; *Zagotta et al., 1990*). N-type inactivation occurs through a ball-and-chain mechanism wherein the amino-terminal region of the channel plugs the internal pore once it opens in response to membrane depolarization (*Demo and Yellen, 1991*; *Hoshi et al., 1990*; *Venkataraman et al., 2014*; *Zagotta et al., 1990*; *Zhou et al., 2001a2001*). However, Shaker channels lacking this N-terminal region can still inactivate on a slower timescale (sec) through a process termed slow or C-type inactivation (*Hoshi et al.,*

*1991*). C-type inactivation regulates the availability of functional Kv channels, modulating the firing of action potentials in neurons and cardiac muscle (*Hoshi and Armstrong, 2013*; *Kurata and Fedida, 2006*). In the human ether-a-go-go related gene (hERG) Kv channel, rapid entry of the channel into the C-type inactivated state underlies the unique inwardly-rectifying properties of the channel that are critical for shaping the cardiac action potential (*Keating and Sanguinetti, 2001*; *Smith et al., 1996*).

Although C-type inactivation in Kv channels is thought to involve a conformational change in the selectivity filter at the external end of the pore (*Armstrong and Hoshi, 2014*; *Choi et al., 1991*; *Hoshi and Armstrong, 2013*; *Hoshi et al., 1991*; *Kurata and Fedida, 2006*; *Liu et al., 1996*; *López-Barneo et al., 1993*; *Perozo et al., 1993*; *Yang et al., 1997, 2002*; *Yellen et al., 1994*), its mechanism remains elusive. In the KcsA K$^+$ channel, a K$^+$ selective channel that lacks voltage-sensing domains and whose pore is related to that domain in Kv channels, X-ray structures of mutants that stabilize open and inactivated states have been solved (*Cuello et al., 2017, 2010*), but the extent to which these inform on the mechanism of slow inactivation in KcsA or Kv channels has been controversial (*Devaraneni et al., 2013*; *Hoshi and Armstrong, 2013*; *Matulef et al., 2013*; *Valiyaveetil, 2017*).

The X-ray crystal structure of Kv1.2, a mammalian homolog of the Shaker Kv channel, and those of the Kv1.2–2.1 chimera containing the S3b-S4 paddle motif from Kv2.1 (Kv chimera), serve as a structural framework for understanding the gating mechanism of Kv channels (*Long et al., 2005a, 2005b, 2007*). The 2.4 Å X-ray structure of the detergent solubilized Kv chimera solved in the presence of lipids has activated voltage sensors, an open internal pore, and a selectivity filter with K$^+$ ions occupying all four ion binding sites, suggesting that it represents an open conducting conformation (*Long et al., 2007*). The presence of an open conducting state in the crystal structure was unanticipated because the Kv chimera C-type inactivates at 0 mV when reconstituted into lipid membranes (*Tao and MacKinnon, 2008*) (*Figure 1—figure supplement 1C*). To characterize the non-conducting C-type inactivated state, Lu and colleagues recently solved the X-ray structure of a non-conducting mutant of the Kv chimera (*Pau et al., 2017*). Their crystals contained two molecules in the asymmetric unit, one of which showed subtle changes to the structure and ion occupancies of the selectivity filter, while the second was indistinguishable from the Kv chimera open conducting state even though this mutant channel is functionally non-conducting. These results suggest that the open and inactivated states are either very similar, or that the open state is preferentially stabilized by some aspect of the crystal environment, such as crystal contacts and/or the detergent and lipid molecules surrounding each channel. Indeed, the gating properties of Kv channels have been shown to be exquisitely sensitive to the lipid composition and mechanical state of the membrane (*Hite et al., 2014*; *Milescu et al., 2009, 2007*; *Ramu et al., 2006*; *Schmidt et al., 2006*; *Schmidt and MacKinnon, 2008*; *Xu et al., 2008*). Since no structure has been obtained for a voltage-activated ion channel in a membrane environment, in the present study we solved the structure of the Kv chimera reconstituted in a lipid nanodisc environment using single-particle cryo-electron microscopy (cryo-EM).

## Results

### Reconstitution of the Kv chimera into lipid nanodiscs

Nanodiscs are hockey-puck shaped lipid bilayers that provide a membrane-like environment for membrane proteins and have been used for structure determination of other membrane proteins using single-particle cryo-EM (*Autzen et al., 2018*; *Chen et al., 2017*; *Dang et al., 2017*; *Efremov et al., 2015*; *Frauenfeld et al., 2011*; *Gao et al., 2016*; *Gatsogiannis et al., 2016*; *Gogol et al., 2013*; *Guo et al., 2017*; *Jin et al., 2017*; *McGoldrick et al., 2018*). A typical nanodisc with an embedded membrane protein is composed of a lipid bilayer containing a membrane protein and two membrane scaffold proteins (MSP) that surround the bilayer (*Denisov and Sligar, 2017*; *Hagn et al., 2013*). We first optimized the assembly condition for empty nanodiscs because the success and the yield of nanodisc assemblies critically depend on the ratio between MSP and lipids used for membrane protein reconstitution (*Grinkova et al., 2010*; *Inagaki et al., 2013*). Using size-exclusion chromatography to assess nanodisc assembly, we found that the assembly of empty nanodiscs was optimal at a 1:100 molar ratio of MSP and lipids, and that using lower or higher ratios of

MSP to lipid resulted in lower yields or formation of large aggregates that elute in the void volume (*Figure 1A*).

We next expressed and purified the α and β subunits of the Kv chimera as previously described (*Long et al., 2005a*, *2005b*, *2007*; *Parcej and Eckhardt-Strelau, 2003*), and found that the purified channel was functional after reconstitution into planar lipid bilayers (*Figure 1—figure supplement 1*). The purified Kv chimera could also be reconstituted into nanodiscs using a 1:10:1000 molar ratio of Kv chimera:MSP:lipid. When assessed with size-exclusion chromatography, two peaks eluting at 10.5 and 12.3 ml were observed along with a void peak (*Figure 1B*, red line). Analysis of these peaks using SDS-PAGE shows that the first corresponds to the Kv chimera reconstituted into nanodiscs and the second to empty nanodiscs (*Figure 1C*). Considering that embedding the Kv chimera into nanodiscs would necessarily displace lipid molecules, we tested a lower ratio of lipid to MSP with a fixed ratio of MSP to Kv chimera, and found that reconstitution of the channel at a ratio of 1:10:400 Kv:MSP:lipid increased the yield of nanodiscs containing the Kv chimera (*Figure 1B*, blue line).

## Electron microscopy of the Kv chimera in lipid nanodiscs

To evaluate the quality of our sample, we used negative staining electron microscopy (*Figure 1—figure supplement 2*). The sample showed nicely distributed particles with easily recognizable side views of the channel harboring a rod-like shape for the transmembrane domain surrounded by a lipid nanodisc, and a cone-shaped cytosolic domain protruding from the nanodisc. Besides the main species composed of the full complex in lipid nanodiscs harboring both α and auxiliary β subunit (α4β4), a small number of complexes missing the β subunits were also identified in the micrographs as well as in individual 2D class averages (α4), indicating some heterogeneity. In addition, the cytosolic domain protruded from the membrane plane at different angles (*Figure 1—figure supplement 2B*), most likely due to the flexible, long linkers connecting the T1 domain in the cytosol and the transmembrane domain of the four α subunits.

Since the Kv chimera sample looked promising, we prepared grids for cryo-EM. After screening different types of grids, micrographs collected on a holey gold grid showed the best results with recognizable nicely distributed particles of the Kv chimera in nanodiscs (*Figure 2A*), and 2D class averages of these particles illustrated views from different orientations with details in the cytosolic as well as the membrane domain (*Figure 2B*). We initially generated a map of the entire complex (α4β4) and obtained an average resolution of the entire map of 3.3 Å. However, the map exhibited a lower local resolution in the transmembrane domain (resolutions of ∼4–7.5 Å; *Figure 2C*), likely

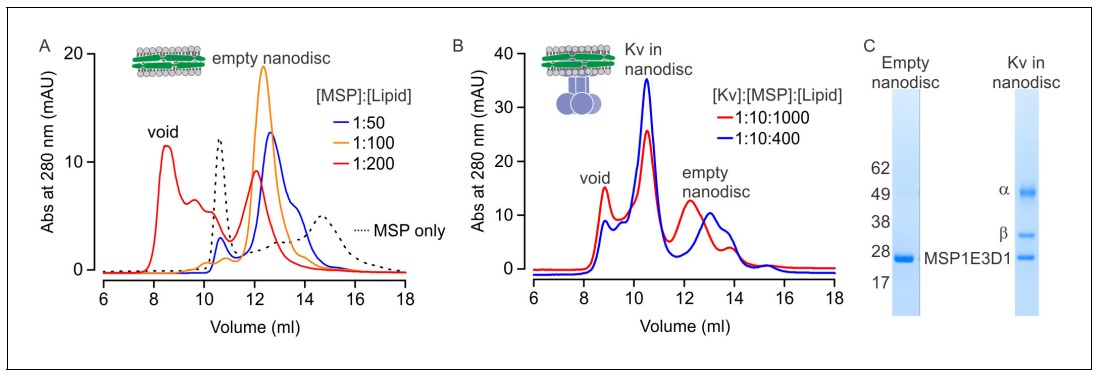

**Figure 1.** Reconstitution of the Kv chimera into lipid nanodiscs. (**A**) Size exclusion chromatograms of empty nanodiscs assembled under different molar ratios of MSP1E3D1:lipid where the lipid mixture is 3:1:1 POPC:POPG:POPE. (**B**) Reconstitution of Kv chimera into nanodiscs. Size exclusion chromatograms of the Kv chimera reconstituted into nanodiscs at different molar ratios of Kv chimera:MSP1E3D1:lipid (same lipids used for empty nanodiscs). (**C**) SDS-PAGE of main fractions collected from size exclusion chromatography (**B**) for nanodisc assembly at molar ratios of 1:10:1000.

DOI: https://doi.org/10.7554/eLife.37558.002

The following figure supplements are available for figure 1:

**Figure supplement 1.** Purification and functional analysis of Kv chimera.

DOI: https://doi.org/10.7554/eLife.37558.003

**Figure supplement 2.** Negative staining electron microscopy of purified Kv chimera in lipid nanodiscs.

DOI: https://doi.org/10.7554/eLife.37558.004

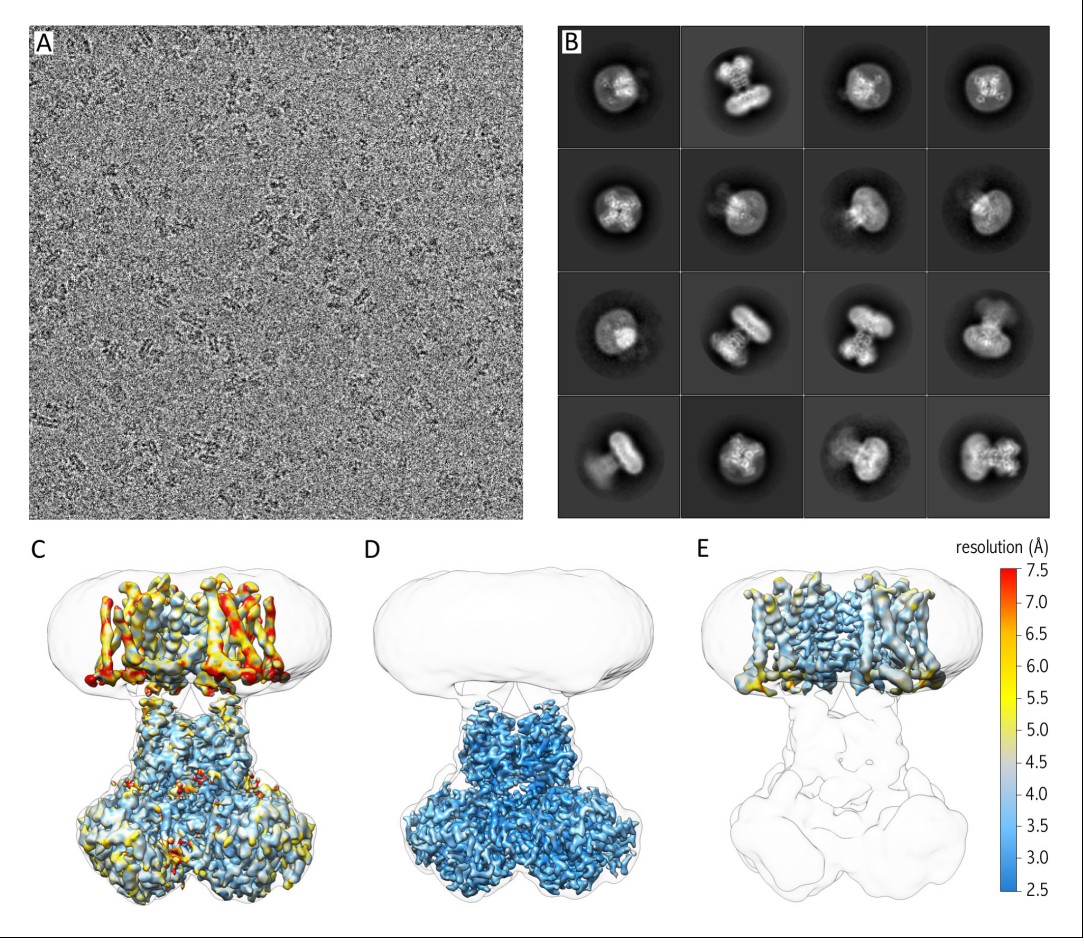

**Figure 2.** Cryo-EM of purified Kv chimera in lipid nanodiscs using gold grids. (**A**) A representative micrograph at approximately −1.7 μm defocus. (**B**) 2D class averages of the 16 most populated classes out of 118,556 selected particles. (**C**) 3.3 Å map of entire complex filtered and colored according to local resolution fitted into a low resolution 3D class envelope. (**D**) 3 Å map of the cytosolic domain after focused refinement colored according to local resolution and fitted into a low resolution 3D class envelope. (**E**) 4 Å map of the transmembrane domain after focused refinement and local resolution filtering, colored according to local resolution and fitted into a low resolution 3D class envelope. FSC curves for each map are shown in *Figure 2—figure supplement 2*.

DOI: https://doi.org/10.7554/eLife.37558.005

The following figure supplements are available for figure 2:

**Figure supplement 1.** Data processing details.
DOI: https://doi.org/10.7554/eLife.37558.006
**Figure supplement 2.** FSC curves indicating overall resolution of cryo-EM maps.
DOI: https://doi.org/10.7554/eLife.37558.007

stemming from the alignment of particles against the cytosolic domain during 3D classification and 3D refinement, and the variability in relative arrangements of the cytosolic and transmembrane domains. To obtain maps with a higher local resolution, we performed particle subtraction, followed by additional classifications as well as 3D refinement on the cytosolic and transmembrane domain independently (*Figure 2D and E*, *Figure 2—figure supplement 1*). The additional two maps showed higher resolution for each domain (3 Å for the cytosolic domain with local higher resolution as high as ~2.5 Å, and ~4 Å for the transmembrane region with local higher resolution of up to ~3.0 Å), which we used to build models of the Kv channel (*Figure 2—figure supplements 1* and *2*; *Table 1*).

**Table 1.**

| | |
|---|---|
| Protein concentration | 0.7 mg/ml |
| Sample volume for EM grid | 3 μl |
| Grid type | UltraAuFoil 1.2/1.3 |
| Plunge freezer | Leica GP EM |
| Blotting time | 10 s |
| Blotting temperature | 4°C |
| Blotting chamber humidity (set/measured) | 95%/88% |
| Microscope | FEI Titan Krios |
| Camera | K2 Summit |
| Energy Filter | no |
| Cs corrector | no |
| Magnification | 29,000x |
| Dose rate | ~4.5 e$^-$/px • s |
| Total dose | ~40 e$^-$/Å$^2$ |
| Number of frames | 38 |
| Physical pixel size (microscope) | 0.858 Å/px |
| Super-resolution pixel size (microscope) | 0.429 Å/px |
| Final pixel size used for final maps | 0.835 Å/px |
| Number of total micrographs | 3,085 |
| Number of selected micrographs | 2,062 |
| Number of particles picked | 281,021 |
| Number of particles selected containing a lipid nanodisc | 118,556 |
| Number of particles in α4β4 map | 47,482 |
| Number of particles in the cytosolic domain map | 57,384 |
| Number of particles in the TM domain map | 65,745 |
| Average resolution in α4β4 map | 3.3 Å |
| Average resolution in cytosolic domain map | 3.0 Å |
| Average resolution in TM domain map | 4.0 Å |
| B-factor for α4β4 map | −89 |
| B-factor for cytosolic domain map | −84 |
| B-factor for TM domain map | −123 |

DOI: https://doi.org/10.7554/eLife.37558.008

## Structure of the Kv chimera in lipid nanodiscs

We built three models for the Kv chimera: one for the cytosolic domain, another for the transmembrane domain, and a third hybrid model that combines the two models which fits in the overall 3.3 Å resolution map of the entire complex (*Figure 3*). For the higher resolution cytosolic domains, cryo-EM densities for most of the side chains were clearly visible (*Figure 3—figure supplement 1*), we could resolve density from the NADP$^+$ cofactor bound to the β subunit (*Figure 3—figure supplement 1G*) and densities for bound water molecules in areas where local resolution was better than 2.8 Å (*Figure 3—figure supplement 1E–F*). Comparison of the cryo-EM structure in nanodiscs and the X-ray crystal structure in detergent shows that they are similar (*Figure 3C*). We do see a slight shift/rotation of the cytosolic region of the α subunit and β subunit with respect to the transmembrane domain of the α subunit when compared to the crystal structure, with the most pronounced changes in the β subunit (*Figure 3A,C,D*), likely reflecting the lack of contacts between molecules that are present in the crystal of the Kv chimera (*Figure 3—figure supplement 2*).

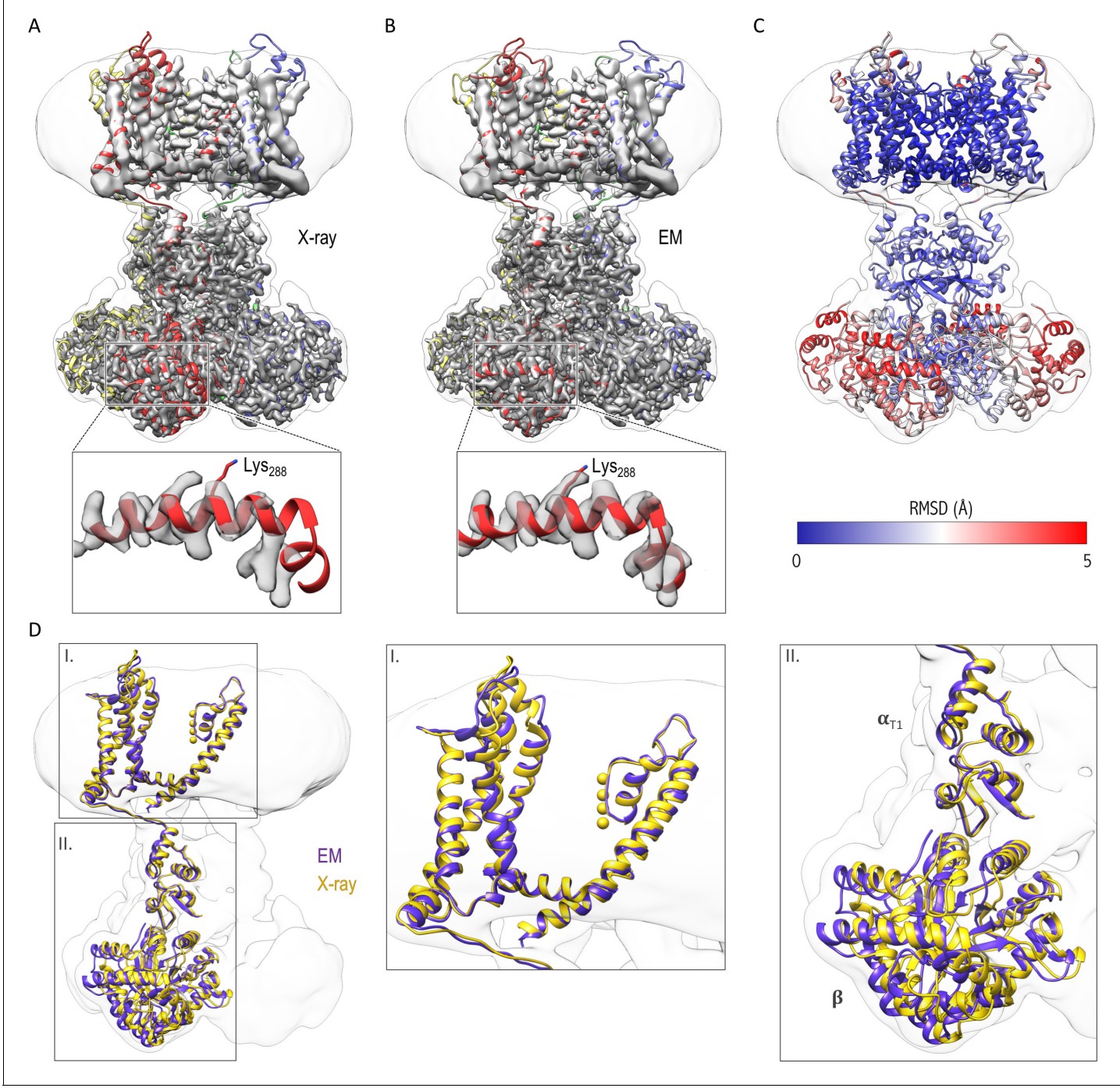

**Figure 3.** Comparison of the structures of the Kv chimera solved in detergent by X-ray crystallography and in lipid nanodiscs by single-particle cryo-EM. (A) The X-ray structure (PDB entry 2r9r) is shown in the overlapping cryo-EM maps with fitting of the transmembrane region to the map. Notice the slight displacement/shift of the X-ray structure is most pronounced in the cytosolic β subunit. (B) Final model refined against the cryo-EM maps is shown with the maps. (C) The structural difference between the X-ray structure and the cryo-EM structure is illustrated with the backbone atom RMSD values (0.096–5.759 Å). α subunits were used for superpositioning of the two structures. (D) The superpositioning of the X-ray (gold) and cryo-EM (purple) structure with the close-up views of the transmembrane (I) and cytosolic (II) domains.

DOI: https://doi.org/10.7554/eLife.37558.009

The following figure supplements are available for figure 3:

**Figure supplement 1.** Quality of the cryo-EM density of the overall 3.0 Å resolution map of the cytosolic domain of the Kv chimera.

DOI: https://doi.org/10.7554/eLife.37558.010

**Figure supplement 2.** Crystal contacts between the Kv chimera β subunits in crystals used to solve the X-ray structure.

*Figure 3 continued on next page*

*Figure 3 continued*

DOI: https://doi.org/10.7554/eLife.37558.011

**Figure supplement 3.** Quality of the cryo-EM density map of the transmembrane domain at an overall 4 Å resolution.

DOI: https://doi.org/10.7554/eLife.37558.012

Although the transmembrane domain exhibits overall lower resolution than the cytosolic domain (*Figure 2*), we see cryo-EM densities for all transmembrane helices (S1-S6) with densities visible for many of the bulky side chains (*Figure 3—figure supplement 3*), which enabled us to refine a model for the entire complex. In addition, the central pore domain shows higher local resolutions (3–4 Å) compared to the peripheral voltage-sensing domains (resolutions of ~4–6 Å), with the lowest resolution for the peripheral loops in the latter domain (*Figure 2E*). Within the pore domain, we observed

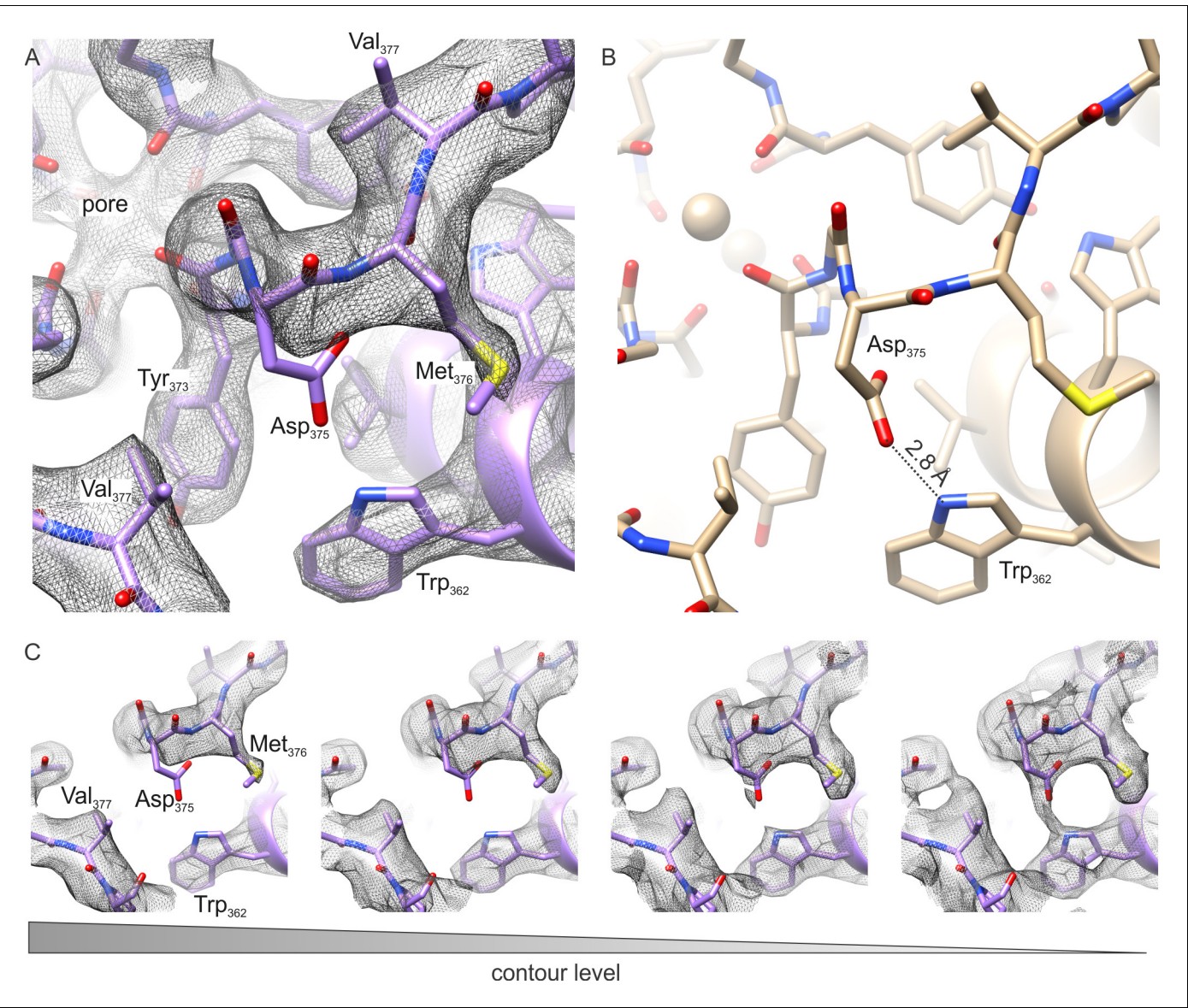

**Figure 4.** The outer pore region of the Kv chimera. (A–B) Extracellular views of the outer pore region of the structures of the Kv chimera solved by cryo-EM (A) and X-ray crystallography (B; 2r9r). D375 and W362, the key residues that are expected to be involved in C-type inactivation are shown. (C) Weak sidechain density for D375 shown at different contour levels.

DOI: https://doi.org/10.7554/eLife.37558.013

densities for most side chains (*Figure 3—figure supplement 3*, *Figure 4*), providing confidence in the structural model. Overall, the transmembrane domain only shows minimal structural differences when compared to the crystal structure (*Figures 3* and *4*), suggesting that the X-ray structure is representative of the structure of the protein in a membrane environment.

We analyzed the cryo-EM map and model within the pore domain to investigate structural changes that may underlie C-type inactivation. For the intracellular end of the pore where the activation gate resides (*del Camino and Yellen, 2001*; *Holmgren et al., 1998*, *Holmgren et al., 1997*; *Webster et al., 2004*), we observed no discernable structural differences between the cryo-EM and X-ray structures (*Figure 3*). The S4-S5 linker that is critical for coupling the voltage-sensing and pore domains is also very similar between cryo-EM and X-ray structures (*Figure 3*). Within the external pore region that has been implicated in C-type inactivation (*Armstrong and Hoshi, 2014*; *Choi et al., 1991*; *Hoshi and Armstrong, 2013*; *Hoshi et al., 1991*; *Liu et al., 1996*; *López-Barneo et al., 1993*; *Yang et al., 1997*; *Yellen et al., 1994*), we see no evidence for significant constriction or dilation of the pore (*Figure 4*, *Figure 5*, *Figure 5—figure supplement 1*).

Although the cryo-EM density maps we have at present of the transmembrane region are at slightly lower resolution when compared to the cytosolic regions, careful inspection of the selectivity filter provides an opportunity to speculate whether the Kv chimera in our experiments is in the C-type inactivated state. Hydrogen bonding between D447 and W434 in the Shaker Kv channel (D375 and W362 in Kv chimera, respectively) has been shown to prevent C-type inactivation (*Pless et al., 2013*). Mutation of these residues profoundly speeds C-type inactivation in the Shaker

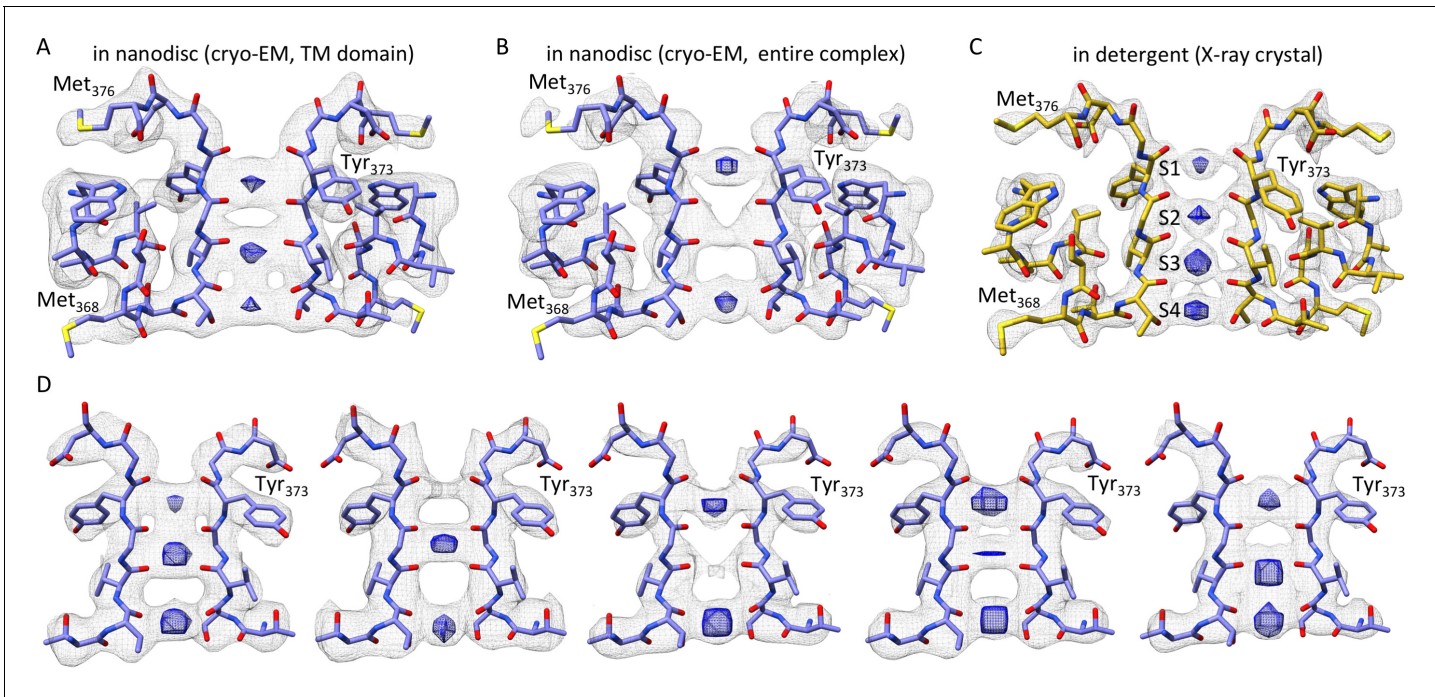

**Figure 5.** The selectivity filter of the Kv chimera. (A) Transmembrane view of the selectivity filter of the Kv chimera solved by cryo-EM in lipid nanodiscs with densities in the pore from the ~4.0 Å map with local higher resolution in this central region of 65,745 signal subtracted particles including only the transmembrane region. (B) Cryo-EM structure of the selectivity filter from the ~3.3 Å map of the entire complex of 47,482 particles. (C) Same view of the selectivity filter of the Kv chimera solved by X-ray crystallography (PDB entry 2r9r). (D) A variety of maps obtained during processing. From left to right, ~3.6 Å map of the entire complex obtained from 67,300 particles, ~3.8 Å map of the entire complex obtained from 75,633 particles, ~3.9 Å map of the entire complex obtained from 63,826 particles, ~3.9 Å map of the entire complex obtained from 118,556 particles, and ~4.6 Å map with local higher resolution in this central region of the transmembrane domain after signal subtraction and focused refinement of 34,654 particles (right). The blue densities are shown from the same maps in gray, just at higher contour level and restricted to the central region of the pore.
DOI: https://doi.org/10.7554/eLife.37558.014

The following figure supplement is available for figure 5:

**Figure supplement 1.** Comparison of the selectivity filters of the Kv chimera and KcsA.
DOI: https://doi.org/10.7554/eLife.37558.015

Kv channel, strengthening the bond by incorporating fluorinated Trp slows C-type inactivation and weakening the bond by incorporating 2-amino-3-indol-1-yl-propionic acid speeds inactivation (*Lueck et al., 2016*; *Perozo et al., 1993*; *Pless et al., 2013*). Mutation of W366 in the Kv1.2 channel, the equivalent of W434 in Shaker, also speed C-type inactivation, albeit more modestly compared to Shaker, and both external $K^+$ and TEA slow C-type inactivation in Kv1.2 (*Cordero-Morales et al., 2011*; *Ishida, 2014*), suggesting that the mechanisms of inactivation are similar between these Kv channels. Evidence for this hydrogen bond is clearly seen in all previous X-ray structures of the Kv chimera (*Banerjee et al., 2013*; *Long et al., 2007*; *Pau et al., 2017*; *Tao et al., 2010*) and KcsA (*Cuello et al., 2017*, *2010*; *Devaraneni et al., 2013*; *Matulef et al., 2013*; *Valiyaveetil, 2017*; *Valiyaveetil et al., 2006*; *Zhou et al., 2001b*) (*Figure 4B*). The structure that we have determined by cryo-EM is also potentially consistent with the presence of H-bonding between D375 and W362 (*Figure 4C*), similar to that reported in the X-ray structure. In this case, it would be necessary to invoke a different mechanism to explain inactivation of the Kv chimera. However, it is also possible that the Asp residue can adopt multiple rotameric positions, and that the loss of this H-bonding interaction may well account for the mechanism of inactivation, as proposed previously (*Pless et al., 2013*). In our structure of the Kv chimera, sidechain density for D375 (equivalent to D447 in Shaker) is weak compared to other nearby residues, including W362, Y373, M376 and V377 (*Figure 4A,C*), raising the possibility that the critical hydrogen bond between D375 in the P-loop and W362 in the pore helix may be broken (*Figure 4A,C*).

A second piece of evidence that the Kv chimera may be C-type inactivated is that we find substoichiometric ion occupancy of the selectivity filter in our maps. We observed that processing different subsets of the data, containing different constellations of particles resulted in density maps where ion densities were absent in different positions (*Figure 5A,B,D*). In the final refined map, which is the one with the highest resolution in the transmembrane region, the ion density is absent in the S2 position (*Figure 5A*). In contrast, in the X-ray structure, all four ion sites exhibit comparable occupancy (*Figure 5C*). The origin of the variations we present in *Figure 5A,B and D* may well be because of the variability in resolution in these maps, and also because of the possibility of artifactual densities along the symmetry axis. Nevertheless, our results suggest that ion interactions with the selectivity filter are likely perturbed for the Kv chimera in a lipid bilayer environment as compared to the environment of the crystal lattice.

## Discussion

The objective of the present study was to determine the structure of the Kv chimera in a membrane environment, both to assess the extent to which previous X-ray structures of detergent solubilized protein capture a native conformation of these channels, and to obtain structural information on the C-type inactivated state. Although the highest resolution X-ray structure of the Kv chimera is at 2.4 Å resolution (*Long et al., 2007*), and our cryo-EM structure is at a resolution of ~3 Å in the cytosolic domain and ~4 Å in the transmembrane domain, our structure in a membrane environment implies that the protein adopts an essentially native structure in detergent in the presence of lipids. We detected only subtle structural changes in the external pore in the selectivity filter of the Kv chimera, as well as a small rotation of the cytosolic region of the α subunit and the β subunit, a region involved in crystal packing in the X-ray structure.

C-type inactivation in Kv channels has been extensively studied, yet the structural basis of this important mechanism has been controversial. Pioneering studies on the Shaker Kv channel point to a critical role of the external pore; increasing the external concentration of $K^+$ ions and externally applied tetraethylammonium (TEA) slow C-type inactivation, and mutants in the external pore have profound effects on the kinetics of C-type inactivation (*Armstrong and Hoshi, 2014*; *Choi et al., 1991*; *Hoshi and Armstrong, 2013*; *Hoshi et al., 1991*; *Kurata and Fedida, 2006*; *Liu et al., 1996*; *López-Barneo et al., 1993*; *Perozo et al., 1993*; *Yang et al., 1997*, *2002*; *Yellen et al., 1994*). In addition, the T449C mutation in the external pore creates a stable $Cd^{2+}$ bridge between subunits that favors the C-type inactivated state (*Yellen et al., 1994*), leading to the proposal that the selectivity filter constricts during C-type inactivation (*Choi et al., 1991*; *Yellen et al., 1994*). It has also been proposed that the external pore dilates during C-type inactivation based on the finding that C-type inactivated Kv channels can conduct $Na^+$ and larger cations when $K^+$ is removed (*Hoshi and Armstrong, 2013*; *Korn and Ikeda, 1995*; *Loboda et al., 2001*). Our cryo-EM structure of the Kv

chimera in nanodiscs suggests that the structural changes during C-type inactivation are likely not large, and would be compatible with very small movements on the scale <2 Å. Although the resolution of our cryo-EM structure limits the extent to which we could unambiguously identify the structural changes that occur with C-type inactivation, it remains possible that our findings are compatible with earlier suggestions that inactivation involves the breakage of a critical hydrogen bond between D375 in the P-loop and W362 in the pore helix. The functional evidence for a critical role of this hydrogen bond in stabilizing a conducting state of the filter includes both gain- and loss-of-function experiments employing conventional and unnatural amino acid mutagenesis (*Pless et al., 2013*). A recent X-ray structure of a non-conducting mutant of the Kv chimera reported changes in occupancy of the S1 site in the filter without disruption of the hydrogen bond networks discussed above (*Pau et al., 2017*), suggesting that there may be multiple structural mechanisms of slow inactivation in the Kv chimera. Recent structures of the hERG Kv channel have raised the possibility of small structural changes underlying inactivation in that channel, including a role for F627, a residue equivalent to Y373 in the Kv chimera (*Wang and MacKinnon, 2017*).

The process of slow inactivation has also been extensively studied in the KcsA channel. There are a number of properties of slow inactivation in KcsA that are similar to what has been observed for C-type inactivation in Kv channels. Slow inactivation in KcsA becomes slower as the concentration of external $K^+$ is raised (*Chakrapani et al., 2007*), and introduction of Cys at Y82, the equivalent of T449 in the Shaker Kv channel, generates a $Cd^{2+}$ bridge that favors the inactivated state (*Raghuraman et al., 2012*). In addition, mutants in the external pore of KcsA either speed (Y82A) or effectively remove (E71A) slow inactivation (*Cuello et al., 2017*). A recent X-ray structure of the rapidly inactivating Y82A mutant of KcsA with the intracellular gate stabilized in the open state suggests that inactivation in this mutant involves a 2.6 Å constriction at G77 within the selectivity filter, leading to disruption of ion binding sites within the filter and water penetration behind the filter (*Cuello et al., 2017*) (*Figure 5—figure supplement 1B*). Other studies have shown that the D-Ala mutant at the G77 position in KcsA can still slow inactivate even though the mutation would be expected to prevent constriction at this position within the filter (*Devaraneni et al., 2013*), suggesting that KcsA can inactivate without constriction at G77 within the filter. The overall message from these studies is that the filter of KcsA can likely undergo multiple types of structural changes that lead to alterations in ion binding to the filter and a decrease in ion conduction. In our cryo-EM structure, it appears that the $C_\alpha$ of G372 in the filter fits better in the cryo-EM density without constriction of the filter (*Figure 5—figure supplement 1A*). It is also interesting that the key hydrogen bond between D375 and W362 that we highlight in the Kv chimera is preserved in all the KcsA structures that have alterations in ion occupancy of the selectivity filter, and mutants of the equivalent Trp in KcsA (W67F) slow rather than speed inactivation (*Cordero-Morales et al., 2011*). Collectively, these results suggest that the detailed mechanisms of inactivation in the Kv chimera and KcsA may be distinct.

The results discussed above for both the Kv chimera and KcsA suggest that the selectivity filters of these channels can be rendered non-conducting by mechanisms that are distinct for different mutants of the two channels, and that the detailed structural mechanism may not be generalizable. It will be important to assess whether higher resolutions can be achieved in the transmembrane domain of the Kv chimera in a membrane environment since these conditions seem to favor the C-type inactivated state without the need for mutations. Our results also suggest that it may be possible to solve structures of voltage-activated ion channels with voltage-sensing domains in a resting state, and with internal pores that are closed, to unravel the mechanisms of voltage sensing and voltage-dependent gating.

## Materials and methods

### Expression of Kv channels in *Pichia pastoris*

The Kv chimera was produced in *Pichia pastoris* as previously described (*Long et al., 2007*; *Parcej and Eckhardt-Strelau, 2003*). Pichia cells were milled using a ball mill (3 cycles of 3 min milling at 25 Hz with 3 min of cooling in liquid nitrogen using a Retsch mixer mill MM400) at cryogenic temperatures and stored at −80℃ until further use. Breaking the cells under cryogenic temperature using a ball mill was critical for obtaining tetrameric Kv channels. Broken cells were suspended in

buffer A (50 mM Tris, 150 mM KCl, 2 mM tris(2-carboxyethyl)phosphine (TCEP), and 10 mM β-mer-captoethanol pH 7.5) supplemented with 0.1 mg/ml of DNaseI and protease inhibitors (1 μg/ml leu-peptin, 1 μg/ml aprotinin, 1 μg/ml pepstatin, 0.1 mg/ml of trypsin inhibitor, 0.624 mg/ml of benzamidine, and 1.5 mM of phenylmethylsulfonyl fluoride) for 2–3 hr at 4°C. The pH of the solution was checked every hour and KOH solution was added to keep the pH neutral. Cell debris were removed by centrifugation at 3000 x g for 10 min. The supernatant was centrifuged for 1 hr at 35,000 rpm (Beckman, Ti 45 rotor) to collect the membranes, which were flash frozen in liquid nitrogen and stored at −80°C until further use.

## Purification of Kv channels expressed in *Pichia pastoris*

To purify the Kv chimera, collected membranes were thawed and resuspended in buffer A with added protease inhibitors as described above using a Dounce homogenizer. Approximately 50 ml of buffer was used for membranes collected from 10 grams of Pichia cell powder. Fresh n-Dodecyl β-D-maltoside (DDM) was added to the resuspended solution to a final concentration of 65 mM, and membrane proteins were extracted for 2 hr at 4°C with gentle stirring on a magnetic stirrer. Insoluble fractions were removed by centrifugation at 35,000 rpm for 1 hr (Beckman, Ti 45 rotor) and the supernatant fraction was incubated with TALON cobalt resin pre-equilibrated with buffer B (buffer A supplemented with 5 mM DDM) overnight. The next day the resin was washed with 10–20 column volumes of buffer C, which is buffer B supplemented with 30 mM imidazole and 0.1 mg/ml lipid mix-tures composed of 3:1:1 mixture of 1-palmitoyl-2-oleoyl-sn-glycero-3-phosphocholine (POPC), 1-pal-mitoyl-2-oleoyl-sn-glycero-3-phospho-(1'-rac-glycerol) (POPG) and 1-palmitoyl-2-oleoyl-sn-glycero-3-phosphoethanolamine (POPE). Bound proteins were eluted with buffer D (buffer B with 0.1 mg/ml lipid mixture and 300 mM imidazole), concentrated using an Amicon 100 kDa ultrafiltration mem-brane and injected into Superdex 200 (10 × 300 mm) column. Loaded proteins were purified using a buffer composed of 20 mM Tris-HCl pH 7.5, 150 mM KCl, 6 mM n-decyl-ß-D-maltoside (DM), 2 mM TCEP, 2 mM DTT, 1 mM EDTA and 0.1 mg/ml of lipid mixtures described above. Tetrameric Kv chi-mera usually eluted after 10–11 ml, which were used for the reconstitution into lipid nanodiscs on the same day of purification. The purification steps described above was performed at 4°C.

## Recordings of the Kv chimera in a planar lipid bilayer

Planar lipid bilayer experiments were performed as previously described (*Hite et al., 2014*; *Tao and MacKinnon, 2008*). Briefly, freshly prepared Kv chimera channels were reconstituted into lipid vesicles (POPE:POPG 3:1 mixture by mole) at a nominal protein to lipid concentration of 1:10 (by mass). The final vesicle concentration was 10 mg/ml in a buffer containing 450 mM KCl, 10 mM HEPES-KOH (pH 7.5) and 2 mM DTT. Bilayer experiments were performed using a commercial 1 ml polystyrene cup chamber with a 200 μm aperture (Warner Instruments). After filling the front and rear (cup) chambers with asymmetric solutions (150 mM KCl, 10 mM HEPES-KOH pH 7.5 in the front; 15 mM KCl, 10 mM HEPES-KOH pH 7.5 in the rear), a planar lipid bilayer was formed by touching a glass rod dipped in a DPhPC/decane solution (20 mg/ml) against the aperture. 2 μl of the vesicle solution was then added to the front chamber and after waiting several minutes for vesicle fusion, the osmotic and [K$^+$] gradient across the membrane was dissipated by adding 135 mM of KCl to the rear chamber. The membrane was then held at −110 mV (rear chamber voltage relative to front chamber) and channel activation probed with 100 msec depolarization steps every 5 s.

## Reconstitution of Kv channel into nanodiscs

MSP1E3D1 was purified as previously described (*Grinkova et al., 2010*; *Inagaki et al., 2013*). Lipids solubilized in chloroform (Avanti) were dried under the stream of nitrogen and placed under a desic-cator overnight protected from light to completely remove residual chloroform. Dry lipids were resuspended in water thoroughly at a concentration of 20 mg/ml and formed vesicles were sonicated until the solution became nearly transparent. DM was added to a concentration of 26 mg/ml and divided into smaller aliquots which were kept frozen at −80°C until further use. To begin nanodisc assembly, the lipid mixture solubilized by DM was mixed with the Kv chimera puri-fied using size exclusion chromatography in DM (usually 1–2 mg/ml after concentrating the fractions collected from the size exclusion purification) and MSP at room temperature for 30 min. The typical reconstitution concentration of the Kv chimera, MSP and lipid mixture was 3 μM, 30 μM and 12 mM,

respectively. The protein concentrations were calculated by measuring the absorbance at 280 nm using molar extinction coefficients of 454,720 $M^{-1}cm^{-1}$ and 26,930 $M^{-1}cm^{-1}$ for the Kv chimera and MSP1E3D1, respectively. The solution was then transferred to a tube containing Biobeads (30 fold of detergent weight) that were previously washed with methanol and equilibrated in 20 mM Tris-HCl pH 7.5 and 150 mM KCl buffer to initiate nanodisc assembly and mixed at room temperature for 2–3 hr. Biobeads were separated by filtering through a 0.22 μm membrane and the filtrate was injected into Superdex 200 (10 × 300 mm) column using a buffer composed of 20 mM Tris-HCl pH 7.5 and 150 mM KCl for samples used in negative staining and 20 mM HEPES-KOH pH 7.5, 150 mM KCl and 2 mM TCEP for samples used in cryo-EM at 4°C. Using fresh Kv chimera (within 1–2 hr from the collection of fractions from size exclusion purification) for nanodisc assembly was critical for reconstitution into nanodisc.

### Negative staining electron microscopy

The Kv chimera sample in lipid nanodiscs was first diluted to ~0.03 mg/ml and then adsorbed to a freshly plasma cleaned carbon-film grid for 30 s, and stained with 0.7% uranyl formate. Images were collected on an FEI Tecnai T12 equipped with a 4k × 4 k Gatan Ultrascan CCD camera. Image analysis, particle selection, 2D class averaging, and initial 3D modelling were performed using EMAN2 (*Tang et al., 2007*).

### Cryo electron microscopy sample preparation

We initially prepared vitrified specimens by adding 3 μl of Kv chimera at various protein concentrations (in 20 mM HEPES-KOH pH 7.5, 150 mM KCl and 2 mM TCEP) to plasma cleaned Cu R1.2/1.3 holey carbon grids (Quantifoil Micro Tools GmbH, Großlöbichau, Germany); however, either the particles were too crowded at a higher protein concentration or only particles that did not look like the entire Kv chimera were detected at a lower protein concentration. To mitigate this problem, we prepared specimens using grids with a 2 nm carbon layer at a protein concentration of 0.1 mg/ml. Data collected on thin carbon layer grids yielded good 2D classes, but only resulted in low resolution 3D maps. Finally, we prepared specimens by adding 3 μl of Kv chimera at 0.7 mg/ml to a 15 s plasma cleaned 300 mesh UltrAuFoil R1.2/1.3 holey gold support grid (Quantifoil Micro Tools GmbH, Großlöbichau, Germany) (*Russo and Passmore, 2014*, *2016*), and nicely dispersed particles with the shape of Kv chimera were detected in thin ice holes. The gold grid used for data collection was blotted for 10 s after 10 s pre-blotting time, then plunge-frozen in liquid ethane using a Leica EM GP instrument (Leica Microsystems Inc., Buffalo Grove, IL), with the chamber maintained at 4°C and 88% humidity.

### Data collection

Cryo-EM imaging was performed on an FEI Titan Krios microscope (FEI Company) operated at 300 kV, aligned for parallel illumination. Data collection images (*Figure 2A*) were acquired with a K2 Summit camera operated in super-resolution counting mode with a calibrated physical pixel size of 0.858 Å and a super-resolution pixel size of 0.429 Å with a defocus range between −1.0 and −2.5 μm using the Latitude S software version 3.21.1253 (Gatan Inc., Pleasanton, CA). Focusing improved by using the edge of the previously imaged hole on the gold foil grid. No energy filter or $C_s$ corrector was installed on the microscope. The dose rate used was ~4.5 e⁻/pixel·s to ensure operation in the linear range of the detector. The total exposure time was 15.2 s, and intermediate frames were recorded every 0.4 s giving an accumulated dose of ~40 e⁻/Å² and a total of 38 frames per image. A total of 3,085 images were collected over six days. 2,928 and 1,852 images were collected using the same setup on Cu carbon grids and Cu carbon grids with a 2 nm carbon layer, respectively.

### Data processing

Movie frame alignment and CTF determination were done as described previously (*Bartesaghi et al., 2014*). Drift analysis comparing data collected on standard Quantifoil R 1.2/1.3 copper grids, R 1.2/1.3 copper grids with a 2 nm carbon layer and UltrAuFoil R 1.2/1.3 gold grids were performed using IMOD (*Kremer et al., 1996*). To quantify the amount of beam induced movement observed for different grid types, we measured the root mean square (RMS) displacement for movies acquired on 300 mesh R 1.2/1.3 QF grids, 200 mesh R 1.2/1.3 QF grids with a 2 nm carbon

layer, and 300 mesh UltraAuFoil R 1.2/1.3 gold grids. Drift trajectories for each movie were computed using the whole frame alignment routines presented in (*Bartesaghi et al., 2014*), and the RMS displacements were plotted as a function of the frame number. For this analysis 2,853, 827, and 2,062 selected images were used for the QF, QF +2 nm carbon, and gold data sets. Compared to the regular QF grids, the movement was reduced significantly when using either the 2 nm carbon layer or the gold grids.

Particle picking was initially done using a nanodisc-embedded TRPV1 template (*Gao et al., 2016*), which was later replaced with a 3D class representing Kv chimera in nanodiscs. 281,021 particles were extracted from 2,062 selected micrographs using a binning factor of 4 (with respect to the super-resolution image size) and a box size of 256 × 256 pixels. 2D class averages (*Figure 2B*) were generated using RELION 2.1 (*Scheres, 2012*). A low-resolution initial model from negative staining (*Figure 1—figure supplement 2*) was used as reference to initially run 3D classification in RELION. Multiple rounds of 2D and 3D classification (*Figure 2—figure supplement 1*) produced an 118,556 particle-containing subset that was used to re-extract using a binning factor of 2 and a box size of 384 × 384 pixels. Only frames 2 to 20 were used for the final rounds of data processing. Additional 2D and 3D classification, 3D refinement and post-processing using RELION 2.1 produced a final map of the entire complex at 3.3 Å resolution from 47,482 particles with C4 symmetry imposed (*Figure 2C*, *Figure 2—figure supplements 1* and *2A*). Using signal subtraction of the transmembrane region from the particle micrographs and only focusing on the soluble domain using the initial subset of 94,131 particles, followed by 3D classification, 3D refinement of 57,384 particles and postprocessing using RELION, we obtained a map at a resolution of ~3.0 Å (*Figure 2D*, *Figure 2—figure supplements 1* and *2B*). The overall 4 Å map of the transmembrane region (*Figure 2E*, *Figure 2—figure supplements 1* and *2C*) was obtained by using signal subtraction of the cytosolic domain from the particle micrographs using the initial subset of 94,131 particles, followed by 2D, and then 3D classification, 3D refinement of 65,745 particles and postprocessing using RELION. Local resolution was determined using RESMAP 1.1.4 (*Kucukelbir et al., 2014*) and local resolution filtering was performed using RELION 2.1.

## Model fitting

A tetrameric model for the Kv chimera was produced from an X-ray structure (PDB entry 2r9r) and used for rigid body fitting in UCSF Chimera (*Pettersen et al., 2004*). The final pixel size of the map was adjusted to be 0.835 Å based on the best fit of the X-ray coordinates of the cytosolic domain into the 3 Å density map. After rigid body fitting of the tetrameric cytosolic and transmembrane domain, initial flexible fitting was performed in COOT (*Emsley et al., 2010*) by manually going through the entire peptide chain of a single α and β subunit before applying the changes to the other three α and β subunits. After a rough fit using COOT, additional real space refinement was performed using Phenix (*Adams et al., 2010*). Models were subjected to an all-atom structure validation using MolProbity (*Chen et al., 2010*). Figures were produced using UCSF Chimera.

## Acknowledgements

We thank Anirban Banerjee, Miguel Holmgren, Joseph Mindell and members of the Swartz and Subramaniam laboratories for helpful discussions. We thank Zhe Lu for sharing the Kv chimera plasmid. We thank Veronica Falconieri for review of figure design. We thank Anirban Banerjee, Pramod Kumar, Chuljin Lee, and Yufeng Zhou for help with expression and purification of the Kv chimera, Ulrich Baxa for assistance with electron microscopy, Edward Twomey and Alexander Sobolevsky for advice on holey gold grids, and Sayaka Inagaki for help with MSPs. This research was supported by the Intramural Research Programs of the National Institute of Neurological Disorders and Stroke, NIH, Bethesda, MD to KJS, by funds from the Center for Cancer Research, National Cancer Institute, NIH, Bethesda, MD to SS and by a grant of Korean Visiting Scientist Training Award Program from the Korea Health Industry Development Institute to CB. This work utilized the computational resources of the NIH HPC Biowulf cluster (http://hpc.nih.gov). The density maps and refined atomic models have been deposited with the Electron Microscopy Data Bank (accession numbers EMD-9024, EMD-9025 and EMD-9026) and the Protein Data Bank (entry codes 6EBK, 6EBL and 6EBM), respectively.

## Additional information

### Competing interests
Kenton Jon Swartz: Reviewing editor, *eLife*. Sriram Subramaniam: Reviewing editor, *eLife*. The other authors declare that no competing interests exist.

### Funding

| Funder | Grant reference number | Author |
|---|---|---|
| Korea Health Industry Development Institute | HI17C2033 | Chanhyung Bae |
| National Cancer Institute | BC010824 | Sriram Subramaniam |
| National Institute of Neurological Disorders and Stroke | NS002945 | Kenton Jon Swartz |

The funders had no role in study design, data collection and interpretation, or the decision to submit the work for publication.

### Author contributions
Doreen Matthies, Chanhyung Bae, Conceptualization, Data curation, Formal analysis, Validation, Investigation, Visualization, Methodology, Writing—original draft, Writing—review and editing; Gilman ES Toombes, Data curation, Formal analysis, Investigation, Writing—original draft, Writing—review and editing; Tara Fox, Data curation, Investigation; Alberto Bartesaghi, Data curation, Investigation, Writing—original draft, Writing—review and editing; Sriram Subramaniam, Conceptualization, Supervision, Funding acquisition, Writing—original draft, Project administration, Writing—review and editing; Kenton Jon Swartz, Conceptualization, Supervision, Funding acquisition, Investigation, Writing—original draft, Project administration, Writing—review and editing

### Author ORCIDs
Doreen Matthies (iD) http://orcid.org/0000-0001-9221-4484
Chanhyung Bae (iD) http://orcid.org/0000-0003-3407-2857
Gilman ES Toombes (iD) https://orcid.org/0000-0001-8346-1790
Sriram Subramaniam (iD) http://orcid.org/0000-0003-4231-4115
Kenton Jon Swartz (iD) http://orcid.org/0000-0003-3419-0765

### Decision letter and Author response
Decision letter https://doi.org/10.7554/eLife.37558.030
Author response https://doi.org/10.7554/eLife.37558.031

## Additional files

### Supplementary files
• Transparent reporting form
DOI: https://doi.org/10.7554/eLife.37558.016

### Data availability
The density maps and refined atomic models will be deposited with the Electron Microscopy Data Bank and the Protein Data Bank.

The following datasets were generated:

| Author(s) | Year | Dataset title | Dataset URL | Database, license, and accessibility information |
|---|---|---|---|---|
| Doreen Matthies, Chanhyung Bae, Tara Fox, Alberto | 2018 | KV1.2-KV2.1 paddle chimera channel in lipid nanodiscs | http://www.rcsb.org/structure/6EBK | Publicly available at the RCSB Protein Data Bank (accession |

| | | | | |
|---|---|---|---|---|
| Bartesaghi, Sriram Subramaniam, Kenton Jon Swartz | | | | no. 6EBK) |
| Doreen Matthies, Chanhyung Bae, Tara Fox, Alberto Bartesaghi, Sriram Subramaniam, Kenton Jon Swartz | 2018 | KV1.2-KV2.1 paddle chimera channel in lipid nanodiscs | http://www.ebi.ac.uk/pdbe/entry/emdb/EMD-9024 | Publicly available at the Electron Microscopy Data Bank (accession no. EMD-9024) |
| Doreen Matthies, Chanhyung Bae, Tara Fox, Alberto Bartesaghi, Sriram Subramaniam, Kenton Jon Swartz | 2018 | KV1.2-KV2.1 paddle chimera channel in lipid nanodiscs, cytosolic domain | http://www.ebi.ac.uk/pdbe/entry/emdb/EMD-9025 | Publicly available at the Electron Microscopy Data Bank (accession no. EMD-9025) |
| Doreen Matthies, Chanhyung Bae, Tara Fox, Alberto Bartesaghi, Sriram Subramaniam, Kenton Jon Swartz | 2018 | KV1.2-KV2.1 paddle chimera channel in lipid nanodiscs, transmembrane domain of subunit alpha | http://www.ebi.ac.uk/pdbe/entry/emdb/EMD-9026 | Publicly available at the Electron Microscopy Data Bank (accession no. EMD-9026) |
| Doreen Matthies, Chanhyung Bae, Tara Fox, Alberto Bartesaghi, Sriram Subramaniam, Kenton Jon Swartz | 2018 | KV1.2-KV2.1 paddle chimera channel in lipid nanodiscs, cytosolic domain | http://www.rcsb.org/structure/6EBL | Publicly available at the RCSB Protein Data Bank (accession no. 6EBL) |
| Doreen Matthies, Chanhyung Bae, Tara Fox, Alberto Bartesaghi, Sriram Subramaniam, Kenton Jon Swartz | 2018 | KV1.2-KV2.1 paddle chimera channel in lipid nanodiscs, transmembrane domain of subunit alpha | http://www.rcsb.org/structure/6EBM | Publicly available at the RCSB Protein Data Bank (accession no. 6EBM) |

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
