## [Decision Letter]

Thank you for submitting your article "Single-particle cryo-EM structure of a voltage-activated potassium channel in lipid nanodiscs" for consideration by *eLife*. Your article has been reviewed by three peer reviewers, including Leon D. Islas as the Reviewing Editor and Reviewer #1, and the evaluation has been overseen by Richard Aldrich as the Senior Editor. The following individual involved in review of your submission has agreed to reveal their identity: Fred Sigworth (Reviewer #3).

The reviewers have discussed the reviews with one another and the Reviewing Editor has drafted this decision to help you prepare a revised submission.

Summary:

In this manuscript Matthies et al., present an impressive and high-quality cryo-EM structure of the Kv1.2-2.1 paddle chimaera in nanodiscs, which is one of the few cryo-EM structures of Kv channels and is the first with high resolution. Several of the findings are exciting. The authors have refined their structure to an overall resolution of 3.4 Angstroms (Å), although the resolution of the transmembrane domain is only 4.3 Å. At this resolution, the main structural features of the model are almost indistinguishable from those of the structural model of the same protein obtained from X-ray diffraction, which is a nice indication that the crystallographic structure is not grossly distorted by the lack of a membrane. Although at this resolution it is ambiguous to assign side chains and geometry to the model, the authors find that some features around the pore region can be discerned and they speculate about the possible state of their model in relation to C-type inactivation, which is an inactivated state often found in potassium channels, although not well characterized in Kv1.2. One of the features of C-type inactivation in the Shaker channel is that it is highly sensitive to the state of a network of hydrogen bonds between aromatic residues and other residues, particularly acidic ones, that form a stabilizing cuff around the selectivity filter. Mutating some of these residues (in Shaker) accelerate or decelerate the rate of entry into the C-inactivated state(s). In the model presented in this work, the authors find that a critical H-bond between Trp362 and D375 seems to not be present, the distance between residues being 4.5 Å. They also find that another H-bond between Tyr373 and Trp363 is also not present. These structural features would be consistent with the present structure in nanodiscs being in the C-type inactivated state, although other expected structural features, such as a constriction of the external part of the pore are not present.

Essential revisions:

1) The main concern, shared by all three reviewers, is that the low resolution might not allow the correct assignment of rotamers in all important residues and this might lead to an erroneous conclusion about the state of the structure. It is essential that this issue of fitting residues to the densities is dealt with more carefully.

2) It could also be argued that the apparently long distance from the carboxylate of D375 to W362 (Figure 4) could be due to the phenomenon of "disappearing carboxylates" that has been argued to arise either from radiation damage or negative electrostatic potential. It would be helpful for the authors to address this issue. One possibility might be for the authors to display other nearby carboxylates (to show that they are visible) or perhaps even to show the density and model of a similar H-bonded pair in a nearby location in the map. Meanwhile, do you have any idea what might be the H-bond partner of D375 in your proposed model?

3) Similarly, the argument that the Y373-W363 H-bond is absent (Figure 4C) depends crucially on the rotameric state chosen for W363. How sure are you of this choice? What about S367, is it also displaced from these residues and not participating in H-bonding?

4) Another point to address is that it is important to show experimentally (or through citations of literature) that perturbations of these interactions do indeed affect C-type inactivation in the chimera channel as the results quoted are for the Shaker K^+^ channel. There may be differences in the effect of mutations on inactivation in Shaker and Kv1.2 chimeric channel. In this regard, the authors might find interesting a recent preprint: (https://www.biorxiv.org/content/early/2018/05/07/316174) and the abstract: (Gating Properties and Voltage Sensing in Kv1.2 Biophysical Journal, Vol. 102, Issue 3, p531a-532a)

---

## [Author Response]

1) The main concern, shared by all three reviewers, is that the low resolution might not allow the correct assignment of rotamers in all important residues and this might lead to an erroneous conclusion about the state of the structure. It is essential that this issue of fitting residues to the densities is dealt with more carefully.

Based on the reviewer’s concerns, we have decided on a more cautious interpretation of the structural changes occurring during C-type inactivation. While our manuscript was in review, we also improved the resolution of our cryo-EM maps using only the earlier frames. The new maps have a resolution of ~4 Å in the TM domain and ~3 Å in the cytosolic domain with local higher resolutions in each domain. As we explain in the text, the structure that we have determined by cryo-EM could well be consistent with the presence of H-bonding between D447 and W434, similar to that reported in the X-ray structure. In this case, it would be necessary to invoke a different mechanism to explain the inactivation of this channel. However, it is certainly possible that the Asp residue can adopt multiple rotameric positions, and that the loss of this H-bonding interaction may account for the mechanism of inactivation, as proposed previously. Based on the biochemical nature of the complexes used for structural analysis, we hypothesize that the Kv chimera is likely to be inactivated because it is in a membrane-like environment at 0 mV and we see evidence of diminished occupancy of ions in the selectivity filter, but whether the hydrogen bond between W362 and D375 is broken or intact remains an open question. We have revised the text accordingly.

2) It could also be argued that the apparently long distance from the carboxylate of D375 to W362 (Figure 4) could be due to the phenomenon of "disappearing carboxylates" that has been argued to arise either from radiation damage or negative electrostatic potential. It would be helpful for the authors to address this issue. One possibility might be for the authors to display other nearby carboxylates (to show that they are visible) or perhaps even to show the density and model of a similar H-bonded pair in a nearby location in the map. Meanwhile, do you have any idea what might be the H-bond partner of D375 in your proposed model?

These are good points but unfortunately, we cannot provide a definitive answer to the question of radiation damage. Based on studies on β-galactosidase, the Subramaniam laboratory reported that there was preferential diminution of density for carboxylate residues at higher electron doses (Bartesaghi et al., 2014), and more recently showed that this reduction can be mitigated by use of proper dose-weighting across all frames (Bartesaghi et al., 2018). Definitive results with β-galactosidase required resolutions better than 2 Å, and our expectation is that once we achieve higher resolutions, we can address questions about H-bonding with greater certainty. Thus, while weaker side chain densities for acidic residues may indeed arise from the absence of hydrogen bonding, the more limited resolution within the TM domain precludes an unambiguous conclusion.

3) Similarly, the argument that the Y373-W363 H-bond is absent (Figure 4C) depends crucially on the rotameric state chosen for W363. How sure are you of this choice? What about S367, is it also displaced from these residues and not participating in H-bonding?

We have reconsidered our choice of rotamer for W363 in light of the reviewers concerns, and decided to choose the same rotamer as is seen in the X-ray structure because it positions the indole nitrogen within the strongest density, and this then leaves that hydrogen bond with Tyr373 intact.

4) Another point to address is that it is important to show experimentally (or through citations of literature) that perturbations of these interactions do indeed affect C-type inactivation in the chimera channel as the results quoted are for the Shaker K^+^ channel. There may be differences in the effect of mutations on inactivation in Shaker and Kv1.2 chimeric channel. In this regard, the authors might find interesting a recent preprint: (https://www.biorxiv.org/content/early/2018/05/07/316174) and the abstract: (Gating Properties and Voltage Sensing in Kv1.2 Biophysical Journal, Vol. 102, Issue 3, p531a-532a)

We now comment on this point and cite two publications, Cordero-Morales, (2011) and the doctoral thesis of Itzel González Ishida where the W366F mutation was studied in the Kv1.2 channel. Both report that the Kv1.2 W366F mutant exhibits slower C-type inactivation when compared to Shaker W434F, however, the mutant still dramatically speeds C-type inactivation compared to wt. In addition, Ishida showed that inactivation in Kv1.2 is slowed by raising external K^+^ or by TEA, features that are qualitatively similar to in Shaker.